# ONC201 for Glioma Treatment: Adding an Important Weapon to Our Arsenal

**Athina-Maria Aloizou** [1,*] and **Dimitra Aloizou** [2]

1   Department of Neurology, St. Josef-Hospital, Ruhr University Bochum, 44892 Bochum, Germany
2   Department of Nursing, National and Kapodistrian University of Athens, 11527 Athens, Greece
*   Correspondence: aaloizou@med.uth.gr; Tel.: +49-172-242-3306

**Abstract:** Glioma, specifically gliobastoma, represents the commonest central nervous system malignancy and is notoriously challenging to treat, with only a minimal number of patients surviving beyond a year after diagnosis. The available treatment options include surgical resection, radiotherapy, and chemotherapy, mainly with temozolomide. However, gliomas can be particularly treatment resistant and novel options are currently being researched. One such agent is ONC201, the first member of the imipridone class and a TNF-related apoptosis inducing ligand (TRAIL)-inducing compound, which has shown positive results in the first preliminary clinical reports about its application in glioma patients, while also being safe and well-tolerated. Particular promise has been shown for the H3K27M mutated glioblastomas, with more trials focusing on this patient subset. It is likely that this compound will be added in the treatment algorithms of glioma in the future, although more research is still needed.

**Keywords:** glioma; glioblastoma; imipridone; ONC201; TRAIL





## 1. Introduction

Glioma, with glioblastoma (GBM) representing the most aggressive stage of the disease, is the commonest primary brain tumor in adults. The mainstays of glioma treatment include surgical resection, radiation, and chemotherapy with temozolomide (TMZ) [1]. GBM poses a therapeutic challenge, as surgical removal may not be feasible due to tumor location, while chemotherapy and radiotherapy often fail due to tumor resistance [1]. Several mechanisms have also been involved in therapy resistance [1], and particular cells within the tumors can lead to recurrence after treatment [2].

TNF-related inducing ligand (TRAIL) belongs to the TNF (Tumor Necrosis Factor) superfamily, which triggers apoptosis via binding to death receptors D4 and D5, and its activation leads to a selective apoptosis of tumor cells [3]. ONC201 is the first member of the imipridone class and has been shown to carry potent anti-cancer properties, by inducing TRAIL and promoting cancer cell death [4]. It selectively targets Dopamine Receptor D2 (DRD2) and ClpP, and upregulates apoptotic factors such as TRAIL. The induction of TRAIL also occurs via inactivation of the Akt and ERK pathways, which in turn leads to the nuclear translocation of transcription factor Foxo3a and the increase of the TRAIL-gene transcription, irrespective of p53 status [5].

ONC201 is administered orally and has shown good tolerability, carrying little to no toxicity to normal cells [6]. The first preclinical studies of this molecule showcased its efficiency against glioma, as well as in tumor cells of patients with TMZ, bevacizumab, and radiation resistance [6], and so in the last few years attempts to use this agent in treating glioma have been made. In this concise review, we attempt to present the available clinical studies of this compound and discuss its application in glioma treatment.

## 2. Clinical Studies of ONC201 in Glioma

The first clinical study (Phase I) was conducted in 2017 and included 10 heavily pretreated patients with refractory solid tumors, and an additional 18 additional patients with advanced disease; no glioma patients were included [7]. This study demonstrated that the compound was well-tolerated and set the recommended dosage at 625mg orally, once every 3 weeks.

The first phase 2 trial (NCT02525692) with GBM came in the same year by Arrillaga-Romany et al., who included 17 patients with recurrent GBM, after surgery, radiation, or TMZ, that received ONC201 652mg every 3 weeks [8]. The median overall survival (OS) was 41.6 weeks, while the 6-month progression-free survival (PFS) was 11.8%, with two patients remaining under the regimen by the end of the study. A 22-year-old patient with a GBM deriving from H3.3 K27M mutant Grade III astrocytoma achieved a partial response within seven cycles, which was maintained for more than 6 months, with her lesions continuously regressing in size. Another patient that received surgery and was put on the medication 7 weeks later was reported as disease-free for more than 11 months. Only two adverse events were noted (one neutropenia and one mild allergic reaction), both of which did not lead to discontinuation of the therapy.

The H3K27M mutated GBM represents a particular type of GBM, commonly presenting as diffuse midline glioma in pediatric patients, and is associated with a very poor prognosis, with TMZ rarely providing any benefit and surgical resection often being impossible due to their localization [9]. The mutation also accounts for >75% of pediatric diffuse intrinsic pontine gliomas, and for approximately half of other midline gliomas in young adults [10]. Drawing on the observations of the first study, especially of the H3K27M mutant patient, Chi et al. (2019) published their trial/case-series of pediatric (11 patients) and young adult (7 patients) H3K27M mutated GBM patients [11], who received ONC201 625mg once weekly. Fourteen patients had recurrent disease, with all patients receiving prior radiation. For these 14 patients, the PFS was 15 weeks for the 7 adults and 13 weeks for the 7 pediatric patients. Five of the patients were still continuing the treatment by the time the trial was terminated, being progression-free, while the remaining 13 discontinued the therapy due to disease progression or death. A heavily treated adult patient with a GBM evolving from a Grade III astrocytoma showed complete response after 11 months with ONC201, while a second adult patient only achieved clinical stability and improvement upon receiving ONC201. Other patients, pediatric as well, reported improvements in their symptoms and daily activities under ONC201 treatment. The first patient of this case-series was also published as a case-report [12], before the inclusion of other patients to the study.

In continuation of the aforementioned phase 2 trial, Arrillaga-Romany et al. (2020) presented the results of their analysis regarding H3K27M adult patients enrolled in ONC201 trials (NCT02525692, NCT03295396, expanded access protocols), where ONC201 was a monotherapy [13]. The compound was once again found to be safe, with 30% of patients showing a good response per response assessment in neuro-oncology (RANO) criteria.

Duchatel et al. (2021) attempted to confirm the effectiveness of a German-sourced ONC201 compound in 28 H3K27M GBM patients, as the compound administered in the precedent studies was not available for Europe [14]. The compound exhibited no in vitro and in vivo efficacy and drug bioavailability differences to the one used in the other trials. The overall survival rates were also comparable to the precedent reports (median OS: 18 months), and were significantly longer compared to historical records (9.2 months, $p = 0.0007$ [15]), although patients were also pretreated with various compounds and regimens.

A phase I dose escalation and expansion trial for ONC201 as a single agent in pediatric diffuse midline gliomas was recently published by Gardner et al. (2022) (NCT03416530) [16]. Twenty-two pediatric patients with diffuse midline gliomas following radiation were enrolled, with the recommended phase 2 dose for adults (625 mg), scaled by bodyweight, set as the primary endpoint. It was achieved, with no dose-limiting adverse effects occurring. Five patients were still alive after 2 years from ONC201 initiation.

As part of the expanded access NCT03134131 trial, Ekhator et al. (2022) also recently published their experience with three H3K27M mutated GBM patients [17]. No serious adverse effects were noted, while the patients exhibited clinical stability and improvements in their symptoms.

## 3. Discussion

TRAIL has been considered an attractive target for oncological treatment strategies due to its ability to selectively target malignant and not normal cells, while its induction also sensitizes tumor cells to chemotherapeutical agents. However, many tumor types exhibit resistance against its action [3], possibly due to epigenetic mechanisms such as microRNA dysregulation [1], and upregulation of TRAIL via agents such as ONC201 may be crucial in efficiently targeting tumor cells.

Upon initial reports of phenomenal response in some patients carrying the H3K27M mutation, the following studies have focused on including patients with this GBM subtype. The H3K27M mutation leads to increased oncogenic transcription via hyperacetylation of the of the wild-type histone H3 variants, and attempts to intervene in this epigenetic disruption have shown some effectiveness in preclinical models [10]. DRD receptors, particularly DRD2, have been revealed as increased in many cancer types, high grade glioma included [18], while it has also been shown that H3K27M mutant midline gliomas are particularly enriched in DRD2 receptors, more so than wild-type gliomas [12,19]. This could possibly explain the efficacy manifested by ONC201, although the compound could provide benefits to patients with wild-type gliomas as well, and it should be tested in the whole glioma spectrum, especially when it has been shown to be a safe drug.

Addressing the role of the H3K27M mutation in overall prognosis is also important. The diffuse midline glioma, H3K27Mmutant, has now been introduced as a new entity in the latest WHO glioma classification, with the mutation being associated with high-grade glioma and reduced overall survival and event-free survival [20], especially for patients <35 years of age [21]. Additionally, two genes, those of O6-methylguanine-DNA-methyltransferase (MGMT) and isocitrate dehydrogenase 1 (IDH-1), have been identified as candidate genes for predicting prognosis and survival in glioma, and have also shown links to H3K27M. In short, an epigenetic silencing (methylation) of MGMT is associated with better response to alkylating agents such as TMZ and better overall survival [22], while IDH-1 mutations are commonly encountered in diffuse gliomas of Grade II and III and secondary GBM, not primary GBM [23]. It has been shown that unmethylated MGMT leads to TMZ resistance in H3K27M mutated cells [24], while H3K27 mutations exhibits associations with IDH1-R132H mutated oligodendroglioma [25]. As such, the worse prognosis for H3K27M gliomas may also stem from an interplay and co-occurrence of H3K27M with other mutations that make the tumor cells particularly therapy-resistant.

Regarding side-effects and safety, neutropenia was reported in a few patients, mostly pediatric, and was shown to resolve upon withdrawal of the dosage for 1 or 2 weeks [8,11,16]. Other common side-effects included headache, nausea, a mild allergic reaction, and a solitary instance of pharmaceutical fever, none of which led to treatment discontinuation [7,8,11,16,17], and no considerable differences regarding these side-effects was noted between adult and pediatric patients.

Of course, the data remains limited. The available reports and trials are few and the patient numbers remain small. Potential differences between adult and pediatric patients also need to be addressed, since differences in the biological backgrounds of gliomas in adults and children have been documented [26]. The results of the available studies do not suffice to support the existence of a divergence in terms of response to ONC201, although a separate examination of these two age groups is warranted. Furthermore, translating the data from the application of ONC201 in other types of malignancies can also prove tricky, since its mechanism of action differs between solid and hematological malignancies, where ONC201 induces p53-independent and caspase-dependent apoptosis, and not TRAIL [27].

However, given how gliomas and GBM in particular are considered deadly and treatment options only show limited effectiveness, it is essential that more people take part in trials assessing this compound, in order for its potential to be confirmed. Although the results so far have not shown a ubiquitous response to this medication, reports of patients achieving remission, clinical stability for years, and improvement of their symptoms only show that it merits more research given the overall unfavorable prognosis of the disease. Additionally, preliminary preclinical evidence has suggested that a combination of ONC201 and other epigenetic modulators may prove more beneficial in the setting of glioma [28], providing ideas for extended research in the future. In this sense, the option of applying ONC201 as a monotherapy in select cases could also be more closely examined, although the effectiveness of this compound in combination and comparison with the mainstays of glioma treatment (resection, radiation, and TMZ) should firstly be assessed. A concomitant application of radiotherapy and the compound might be the best option, with evidence suggesting that ONC201 leads to better radiotherapy results [29]. Improving drug delivery, for example with convection-enhanced delivery (CED), could also provide better therapeutic outcomes, since H3K27M mutated diffuse midline gliomas, where ONC201 has shown promise, cannot be surgically removed, and trials with other chemotherapeutics applied via CED are being conducted [30].

## 4. Conclusions

With a dismal life expectancy of about 15 months after diagnosis in a majority of patients, treating glioma and glioblastoma remains a clinical challenge. The mainstays of therapy in glioma are often proven as ineffective, so novel compounds are more frequently offered to patients in hope of remission. ONC201, a TRAIL-inducing compound, has shown very good tolerability and safety, with numerous reports of patients achieving remission or at least clinical stability under this treatment. Evidence suggests that this compound may be particularly effective for H3K27M mutated tumors, and its application is only expected to grow in the future, although more research is urgently needed.

**Author Contributions:** Conceptualization, A.-M.A. and D.A.; methodology, D.A.; writing—original draft preparation, A.-M.A.; writing—review and editing, D.A. All authors have read and agreed to the published version of the manuscript.

**Funding:** This research received no external funding.

**Institutional Review Board Statement:** Not applicable.

**Informed Consent Statement:** Not applicable.

**Data Availability Statement:** Not applicable.

**Conflicts of Interest:** The authors declare no conflict of interest.

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
