# Peer review of "ONC201 for Glioma Treatment: Adding an Important Weapon to Our Arsenal"

_2571-6980, doi:10.3390/neuroglia4010003_

Round 1

Reviewer 1 Report

This short review by Aloizou and Aloizou addresses the potential anti-glioma effects of ONC201, a new putative anti-cancer drug.  With the present dismal prognosis for glioma even with standard of care of resection, radiation and chemotherapy, newer drugs are highly necessary.  ONC201 is a member of the imipridone class and an active initiator of tumor necrosis factor (TNF)-related apoptosis inducing ligand (TRAIL).  Some clinical studies have shown positive effects in H3K27M mutated diffuse midline gliomas and it remains to be seen whether INC201 can be an effective single agent or an adjuvant anti-glioma drug both in pre-clinical studies and in clinical trials.  This review presents a good summary in that direction.  Some suggestions are given below for the authors' consideration to improve the manuscript. 

Major comments

1. It stresses the safety aspects of the drug than its potential mechanisms with respect to glioma.  It will benefit adding a paragraph on the frequency of H3K27M mutations in relation to the known factors that affect drug efficacy in glioma treatment such as IDH status and MGMT status.

2. The standard of care of resection, radiation and temozolomide needs to be the background on which other putative drugs need to be discussed.

3. Also, the mechanism of ONC201 in hematological malignancies and solid cancers is different.  Therefore, data from the former cannot be extended for effectiveness in the latter.  Such limitations of present data bases would also help in strengthening the review.

Minor comments

1. One minor concern is that language can be improved in some places.  For instance, “Treating glioma and glioblastoma remains a clinical challenge, with the majority of patients expiring soon after diagnosis” can be reworded as “With a dismal life expectancy of about 15 months after diagnosis in a majority of patients, treating glioma and glioblastoma remains a clinical challenge”.          

Author Response

Reply to the Reviewers’ Comments

We would like to thank the Reviewers for taking the time to thoroughly assess our manuscript and suggest improvements. Below we have responded to the commentaries individually and hope that the manuscript is now suited for publication.

Reviewer 1:

This short review by Aloizou and Aloizou addresses the potential anti-glioma effects of ONC201, a new putative anti-cancer drug.  With the present dismal prognosis for glioma even with standard of care of resection, radiation and chemotherapy, newer drugs are highly necessary.  ONC201 is a member of the imipridone class and an active initiator of tumor necrosis factor (TNF)-related apoptosis inducing ligand (TRAIL).  Some clinical studies have shown positive effects in H3K27M mutated diffuse midline gliomas and it remains to be seen whether INC201 can be an effective single agent or an adjuvant anti-glioma drug both in pre-clinical studies and in clinical trials.  This review presents a good summary in that direction.  Some suggestions are given below for the authors' consideration to improve the manuscript.

Major comments

  1. It stresses the safety aspects of the drug than its potential mechanisms with respect to glioma. It will benefit adding a paragraph on the frequency of H3K27M mutations in relation to the known factors that affect drug efficacy in glioma treatment such as IDH status and MGMT status.

Reply: We thank the Reviewer for raising this point. We have added a paragraph in the discussion sections, which now reads as “Addressing the role of the H3K27M mutation in overall prognosis is also important. The diffuse midline glioma, H3K27M–mutant has now been introduced as a new entity in the latest WHO glioma classification, with the mutation being associated with high-grade glioma and reduced overall survival and event-free survival (18), especially for patients <35 years of age (19). Additionally, two genes, those of O6-methylguanine-DNA-methyltransferase (MGMT) and isocitrate dehydrogenase 1 (IDH-1) have been identified as candidate genes for predicting prognosis and survival in glioma, and have also shown links to H3K27M. In short, an epigenetic silencing (methylation) of MGMT is associated with better response to alkylating agents such as TMZ and better overall survival (20), while IDH-1 mutations are commonly encountered in diffuse gliomas of Grade II and III and secondary GBM, not primary GBM (21). It has been shown, that unmethylated MGMT led to TMZ resistance in H3K27M mutated cells (22), while H3K27 mutations exhibited associations with IDH1-R132H mutated oligodendroglioma (23). As such, the worse prognosis for H3K27M gliomas may also stem from an interplay and co-occurrence of H3K27M with other mutations that make the tumor cells particularly therapy-resistant.”

  1. The standard of care of resection, radiation and temozolomide needs to be the background on which other putative drugs need to be discussed.

Reply: We agree with the Reviewer and have now attempted to make this clearer throughout our manuscript. We have also added a relevant statement in the introduction section, which reads “The mainstays of glioma treatment include surgical resection, radiation, and chemotherapy with temozolomide (TMZ).”

  1. Also, the mechanism of ONC201 in hematological malignancies and solid cancers is different. Therefore, data from the former cannot be extended for effectiveness in the latter. Such limitations of present data bases would also help in strengthening the review.

Reply: We agree with the Reviewer and have proceeded to address this point in the discussion section, which now reads as  “Of course, the data still remains limited. The available reports and trials are few and the patient numbers remain small. Furthermore, translating the data from the application of ONC201 in other types of malignancies can also prove tricky, since its mechanism of action differs between solid and hematological malignancies, where ONC201 induces p53-independent and caspase-dependent apoptosis, and not TRAIL (24).”

Minor comments

  1. One minor concern is that language can be improved in some places. For instance, “Treating glioma and glioblastoma remains a clinical challenge, with the majority of patients expiring soon after diagnosis” can be reworded as “With a dismal life expectancy of about 15 months after diagnosis in a majority of patients, treating glioma and glioblastoma remains a clinical challenge”.

Reply: The manuscript was extensively reviewed and corrected, the aforementioned rewording was also included.

Author Response

Reply to the Reviewers’ Comments

We would like to thank the Reviewers for taking the time to thoroughly assess our manuscript and suggest improvements. Below we have responded to the commentaries individually and hope that the manuscript is now suited for publication.

Reviewer 2

In the present study titled “ONC201 for Glioma Treatment: Adding an Important Weapon to our Arsenal” the authors present the available clinical studies of ONC201, a small molecule selective antagonist of dopamine receptor DRD2/3 that causes p53-independent apoptosis in tumor cells, and its application in glioma and glioblastoma patients. This is an interesting review since it focuses attention on an interesting compound showing a safety profile that deserves attention, especially for the treatment of such aggressive tumors with a very dismal prognosis as glioma and glioblastoma. It should be considered for publication in Neuroglia pending some concerns that should be addressed.

In the introduction section pathways related to the mechanism of action of ONC201 should be described more in detail. It is important to mention that ONC201 causes the dephosphorylation of Foxo3a by inducing the inhibition of both ERK and AKT. This leads to the dislocation of Foxo3a into the nucleus inducing the upregulation of gene transcription by binding with the TRAIL promoter.

Reply: We thank the Reviewer for this suggestion. We have now included the following statement: “The induction of TRAIL also occurs via inactivation of the Akt and ERK pathways, which in turn lead to the nuclear translocation of transcription factor Foxo3a and the increase of the TRAIL-gene transcription, irrespective of p53 status (5)”.

In the “Clinical Studies of ONC201 in Glioma” section, please specify the reference values for the overall survival (OS) and the progression-free survival (PFS) in pediatric and adult GBMs or gliomas

Reply: We thank the Reviewer for this suggestion. We have included all available OS and PFS in the Clinical Studies section.

The discussion section needs to be improved.

The first part of the discussion talks about TRAIL which has been recognized as a promising target for cancer therapy. Although TRAIL exhibits high antitumor activity, resistance to TRAIL-induced apoptosis in cancer cells has been considered a clinical obstacle to its application. In the second part, however, it is not clear what the relationship between TRAIL and miRNA is. Ref1 refers to a specific miRNA: miRNA-21 that is found upregulated in gliomas. Why the upregulation of miRNA- 21, induced by ONC201, may be crucial in efficiently targeting tumor cells?

Reply: miRNA-21 is upregulated in glioma, not induced by ONC201. This miRNA-21 upregulation, as stated in the cited paper, leads to TRAIL-resistance, and the inhibition of this miRNA was in vivo proven to sensitize cells against TRAIL again. In the statement “the upregulation of this molecule”, we mean TRAIL, not miRNA-21. We have now written it as “the upregulation of TRAIL”, to clarify any confusion.

Please address this point.

- In light of the reported clinical trial results, is there a difference in drug response between pediatric and adult gliomas?

Reply: The data does not suffice to extract a definite conclusion, though a significant difference does not seem to exist. We have added a relevant statement in the discussion section, which reads as “Potential differences between adult and pediatric patients also need to be addressed, since differences in the biological backgrounds of gliomas in adults and children have been documented (25). The results of the available studies do not suffice to support the existence of a divergence in terms of response to ONC201, though a separate examination of these two age groups is warranted.”

- Slightly expand the part on side effects, possibly distinguishing between those most commonly found in pediatric and those in adults

Reply: Besides the neutropenia, which was also noted in an adult patient, no other significant differences could be seen between adults and children. We have added a relevant statement.

Minor points:

- Please, correct some minor typo errors (for example line 78)

- Please correct the numbers: in line 66-67 the numbers are written in letters while in line 69 the number 13 is written as a number

- Line 51, talking about the progression-free survival that was achieved for 2 patients, it is not addressed how many time they remain PFS

Reply: The typographical errors have been corrected, and the numbers have been corrected, we thank the Reviewer for pointing it out. The numbers remain in full-letters only at the beginning of sentences. The PFS at the aforementioned point has now been corrected, also per Reviewer 1’s suggestions.

- Line 86 - speaking overall survival rates were significantly longer than historical records – please report historical record values or add a reference

Reply: We have now provided the reference used by the author themselves for the comparison.
